# The Impact of Intraocular Treatment on Visual Acuity of Patients Diagnosed with Branch Retinal Vein Occlusions

**DOI:** 10.3390/healthcare11101414

**Published:** 2023-05-12

**Authors:** Diana-Maria Darabuş, Mihnea Munteanu, Maria-Alexandra Preda, Olimpiu Ladislau Karancsi, Marius Cristian Șuță

**Affiliations:** 1Department of Ophthalmology, “Victor Babeş” University of Medicine and Pharmacy, 300041 Timișoara, Romania; diana.darabus@umft.ro (D.-M.D.); mihnea.munteanu@umft.ro (M.M.); marius.suta@umft.ro (M.C.Ș.); 2Department of Oral Implantology and Prosthetic Restorations on Implants, “Victor Babes” University of Medicine and Pharmacy, 300041 Timișoara, Romania; olilk@umft.ro

**Keywords:** visual impairment, Ozurdex, retinal ischemia, macular edema, aflibercept, intraocular treatment

## Abstract

Branch retinal vein occlusions are a significant cause of vision loss and present several ophthalmic and systemic risk factors, including age, hypertension, hyperlipidemia and glaucoma. Retinal vein occlusion is the second-most-common retinal vascular disease. This study evaluated the effects of Ozurdex in contrast to a combination therapy with anti-vascular endothelial growth factor (VEGF) and cortisone in treatment-naive branch retinal vein occlusions-macular edema (BRVO-ME) cases, at 4-month and 6-month follow-ups. Thirty eyes were included in the study, which were divided into two groups. The first group consisted of 15 eyes, and each received 1 injection of dexamethasone intravitreal implant Ozurdex (DEX). The second group of 15 eyes received 3 intravitreal injections, the first and second with the anti-vascular endothelial growth factor aflibercept and the third one with 4 mg of triamcinolone acetonide (Vitreal S), spaced at one month. The best corrected visual acuity (BCVA) results suggested that the peak efficacy was at 4 months for both groups, with mean values of 0.5 LogMAR and 0.4 LogMAR. Regarding macular edema, there were no significant changes between the 4- and 6-month follow-up periods, with mean values of 361 μm and 390 μm. Six patients experienced transient raised intraocular pressure at one week after treatment. This study highlights the benefits to visual acuity of the combination of anti-vascular endothelial growth factor and cortisone, which represents a viable solution with similar results to Ozurdex therapy.

## 1. Introduction

Retinal vein occlusion is the second-most-common among the retinal vascular disorders, and it is a major cause of vision loss worldwide [1].

The incidence of retinal vein occlusions varies from 0.2% to 0.8%, and its prevalence in people over 40 years old is 1–2%. The prevalence of branch retinal vein occlusion (BRVO) is four times higher than that of central retinal vein occlusion (CRVO) [2,3].

The diagnosis of BRVO is generally easy to establish due to its characteristic features. BRVO can sometimes be asymptomatic or with visual blurring in the sector of the visual field, corresponding to the area of the affected retina. In macular BRVO, the central vision is disturbed, but it is normal in the periphery. The diagnosis is established by slit lamp examination and fundoscopy in artificial mydriasis. BRVO macular edema (BRVO-ME) is assessed using optical coherence tomography. The fundus examination reveals flame-shaped, dot and blot hemorrhage; soft and hard exudates; and dilated, tortuous veins. In cases of old occlusions, vascular sheathing and venous collaterals are present. Fluorescein angiography is another investigation that can provide valuable information on the extent of macular edema and ischemia [4].

Most BRVOs occur at an arteriovenous crossing; the narrowed vein experiences turbulent blood flow that induces clotting, causing a blockage or occlusion. This leads to increased retinal capillary pressure, which produces transudation into the extracellular space and leads to macular edema and ischemia [5,6].

BRVO may produce numerous complications such as retinal neovascularization, retinal detachment, vitreous hemorrhage, glaucoma and macular edema (ME), which is the most frequent complication and the most important cause of decreased visual acuity [7].

It is considered that the pathogenesis of BRVO is based on the increased production of vascular endothelial growth factor (VEGF), which leads to excessive vascular leakage and contributes to retinal hemorrhages and progressive retinal ischemia, which in turn stimulates increased levels of VEGF. The first-line treatment of acute BRVOs is the use of injections with anti-VEGF agents, but most anti-VEGF agents for BRVO imply frequent administration due to their relatively short period of action. Corticosteroids (dexamethasone and triamcinolone acetonide) work by penetrating cells and blocking the production of VEGF and prostaglandins—substances that are involved in the process of neovascularization and inflammation. The dexamethasone intravitreal implant Ozurdex (DEX) has a longer duration of action of up to 4–6 months [8].

BRVO was first described by Leber in 1877. In some research, age was considered a factor that influences the onset of BRVO compared with CRVO. The higher proportion was after 65 years of age [9].

There are two types of BRVO: major BRVO, in which a major branch retinal vein is occluded, and macular BRVO, wherein one of the macular venules is obstructed. Studies have shown that there is a frequent location of occlusion; in 66% of eyes with BRVO, occlusion of the major branch occurs in the supratemporal quadrant, and in 22–43% of eyes, occlusion of the major branch occurs in the inferotemporal quadrant. The diagnosis of the nasal location of the BRVO is very rare due to the fact that it is asymptomatic and usually discovered by accident or when it produces complications such as bleeding from neovascularization. The etiology of BRVO should be investigated and treated because the cumulative probability of developing a second episode of occlusion in the other eye within 4 years is about 7%. The pathogenesis of BRVO is a result of a combination of three mechanisms: degenerative changes in the vessel wall, compression of the vein at the arteriovenous crossing, and abnormal hematological factors [10].

Mechanical narrowing of the venous lumen at the arteriovenous crossing plays a role in the pathogenesis of BRVO. Venous occlusion appears at the arteriovenous crossing due to the anatomic features of the site and the effects of arteriolar sclerosis. Anatomically, the thin-walled vein lies between the more rigid thick-walled artery and the retina. Because the retinal artery and vein share the same adventitial sheath and due to the narrow venous lumen at the arteriovenous crossing, this setting is conducive to the occlusion process [11].

Regarding the degenerative changes in the vessel wall, Seitz presented research on the clinical histological relationship in one eye with BRVO a few hours after onset. Even though the fundoscopic examination revealed a strongly dilated and tortuous vein distal to the crossing, no blood thrombus obliterating the venous lumen at the crossing was identified. Endothelium and intima-media changes were present. Research suggests that the pathogenesis of BRVO can be explained by the trophic changes in the venous endothelium and intima-media that follow the compression from the overlaying artery. The formation of the thrombus represents a secondary process [12].

Sclerosis of the retinal artery is associated with systemic hypertension, diabetes mellitus, atherosclerosis, and smoking, which are commonly reported in patients with retinal vein occlusions. The rigid artery wall produces a mechanical obstruction of the vein, which leads to a turbulent blood flow, causing damage to the venous endothelium and intima-media. Turbulent blood flow was identified by Christoffersen and Larsen in a study that analyzed the fluorescein angiograms of 250 patients with BRVO [13].

Hematological disorders such as hyperviscosity due to high hematocrit are associated with BRVO. Blood viscosity is mainly dependent upon the hematocrit and plasma fibrinogen. Dysregulation of the thrombosis–fibrinolysis balance is another hypothesis incriminated in the pathogenesis of BRVO. The results of published research are inconsistent, and the role of coagulation factors in the development of retinal vein occlusions remains uncertain [14].

Over time, many therapies have been trialed to treat macular edema caused by retinal vein occlusions. In previous decades, laser photocoagulation was considered the primary treatment for retinal vein occlusions, even though some studies indicated that there was no significant improvement in vision with laser treatment compared with a control group [15,16,17]. On the contrary, many published studies indicate the effectiveness of anti-VEGF agents and cortisone intraocular injections in treating macular edema associated with retinal vein occlusions [18,19,20,21].

Ongoing treatment options for macular edema do not provide permanent solutions due to frequent recurrence or resistance to anti-VEGF therapy in patients, suggesting that other factors are involved in its pathogenesis [22,23]. Treatment possibilities of monotherapy with anti-VEGF agents or corticosteroids are currently and commonly used, but they have limitations and side effects, i.e., repetitive injections because of the short duration of action, risk of cataract formation, endophthalmitis, and increased intraocular pressure [24,25].

The present study evaluated the effects of Ozurdex in contrast to a combination therapy of anti-VEGF and cortisone in treatment-naive BRVO-ME cases at 4- and 6-month follow-ups.

The purpose of the study was to determine the effectiveness of Ozurdex versus a combined intravitreal therapy of anti-VEGF with triamcinolone acetonide in reducing central macular oedema, 4 and 6 months after treatment. The study objective was to show if a combined therapy of cortisone + anti-VEGF has the same effects on visual acuity and macular edema as Ozurdex therapy. IOP was also determined for both groups, as other studies show that cortisone and anti-VEGF therapy increase IOP [26,27].

## 2. Materials and Methods

This was an open-label, prospective, interventional and randomized study with the consecutive enrolment of patients diagnosed with macular edema due to branch retinal vein occlusion. Patients were randomly separated into two groups before treatment started at a 1:1 ratio.

The inclusion criteria were as follows: positive diagnosis of macular edema due to branch retinal vein occlusion, IOP ≤ 20 mmHg, and signed informed consent. The exclusion criteria were as follows: ocular or general infections in the last 6 months, ocular surgery in the last 3 months, and dense cataracts.

A total of 30 patients were included in this study. Their demographic data are presented in Table 1. At baseline, there were no important differences between the studied parameters in the two groups.

Thirty patients diagnosed with BRVO-ME were enrolled. They were divided into two groups. The first group consisted of fifteen eyes and each one received one injection of DEX. The second group of fifteen eyes received three intravitreal injections; the first and the second one were with 2 mg (0.05 mL) anti-VEGF agent aflibercept, and the third one was with 4 mg triamcinolone acetonide (Vitreal S), spaced at one month. The cortisone was extracted pure from the vial and administrated intraocularly. At baseline, all subjects were evaluated from an ocular and general point of view to fulfill the study criteria. Patients were randomly assigned to one of the study arms. Optical coherence tomography (OCT) was used to determine central macular edema, which we defined as a central macular thickness (CMT) >240 μm. A total of 10 eyes from the first group and 11 eyes from the second group were pseudophakic. The best-corrected visual acuity (BCVA) and intraocular pressure (IOP) were evaluated.

All injections were performed in the operating room under sterile conditions. Local anesthetic (0.4% oxybuprocaine) and antibiotic drops were used before and after the injection. Topical disinfection with povidone-iodine was performed, after which the sterile field and the lid speculum were applied. Injections were performed using 30-gauge needles through the inferotemporal pars plana, 4 mm from the limbus. All patients received a protective eye bandage for a few hours after the procedure.

Regarding the statistical analysis of the results, repeated-measures ANOVA and post-hoc Tukey comparisons were used.

## 3. Results

In order to evaluate and compare the impact on visual acuity of one intravitreal dexamethasone implant versus three combined intraocular injections with cortisone and anti-VEGF in patients with macular edema due to branch retinal vein occlusion, we determined and evaluated three parameters: the BCVA; central macular thickness, which suggests the level of macular edema; and the intraocular pressure. A total of 30 patients were included in the study; 50% received only one DEX injection, while the other 50% received two anti-VEGF injections and one cortisone (Vitreal S) in the first 3 months.

The drugs used in this study were: Ozurdex (Allergan Inc. (Dublin, Ireland) dexamethasone intravitreal implant 0.7 mg, in the NOVADUR™ solid polymer drug delivery system); Eylea INN-Aflibercept 40 mg/mL solution (Regeneron Pharmaceuticals, Inc., Tarrytown, NY, USA); and Vitreal S^®^ Triamcinolone acetonide 4.0% ophthalmic suspension (Fidia Farmaceutici S.p.A., Abano Terme PD, Italy).

### 3.1. BCVA Assessment

BCVA is the most important parameter to evaluate the treatment efficacy by functional measurement [28]. All of the patients included in this study had significant differences in BCVA compared with the baseline level at different time points (4 months and 6 months after treatment).

For the Ozurdex group, the mean improvement in BCVA was significant after 4 months (0.540 LogMAR), reaching a mean peak value of 0.473 LogMAR at 6 months. The baseline mean BCVA was 0.8 LogMAR and increased to 0.5 LogMAR and 0.4 LogMAR at 4 and 6 months. For the cortisone + anti-VEGF group, the mean improvement in BCVA was significant after 4 months (0.413 LogMAR), reaching a mean peak value of 0.367 LogMAR at 6 months. The baseline mean BCVA was 0.9 LogMAR and increased to 0.4 LogMAR and 0.3 LogMAR at 4 and 6 months.

Comparing the BCVA values for the two study groups, at 4- and 6-month follow-ups, showed that there were no major differences—not more than one line on the reading chart—in favor of the combined therapy group. All value fluctuations are presented in Table 2.

Assessing the BCVA for the two study groups demonstrated that patients from the cortisone + anti-VEGF group gained one line more on the reading chart than the Ozurdex group. BCVA was determined before the treatment at baseline and 4 and 6 months after the treatment for each group. All data are presented in Table 2 and Figure 1.

Repeated-measures ANOVA was used to compare group mean differences in LogMAR visual acuity by eye, as shown in Table 3, Table 4 and Table 5. The interaction between visual acuity (LogMAR) and treatment is statistically significant (*p* = 0.009); the values are found in Table 4.

### 3.2. Macular Edema Evaluation

Macular edema was determined by measuring the central macular thickness at baseline and 4 and 6 months after treatment, as shown in Table 6 and Figure 2. CMT could be considered as an anatomical value to evaluate ME before and after treatment [29].

For the Ozurdex group, the mean initial CMT was 427 μm (min. 320 μm, max. 570 μm); the 4- and 6-month values improved significantly to 361 μm (min. 240 μm, max. 536 μm) and 347 μm (min. 260 μm, max. 520 μm), respectively. Six months after treatment, Ozurdex treatments reached an important level (dropping from a mean initial CMT of 427 ± 76.8 μm to the lowest mean CMT of 347 ± 97.9 μm). The mean final change in CMT after the treatments was 80 μm. CMT showed significant improvement in the first 4 months, then fluctuated within a stable range. For the cortisone + anti-VEGF group, the mean initial CMT was 471 μm (min. 394 μm, max. 534 μm); the 4- and 6-month values improved significantly to 390 μm (min. 300 μm, max. 490 μm) and 344 μm (min. 268 μm, max. 480 μm), respectively.

Comparing the CMT values for the two study groups, at 4- and 6-month follow-ups, proved that there were no significant differences. At 4 months, a mean value of 31 μm of difference was present between the two groups, and at 4 months the mean difference was less than 3 μm. All values are presented in Table 6.

At the final follow-up visit, in the cortisone + anti-VEGF group, the CMT reached 344 ± 58.0 μm, and the mean final change was 127 μm. None of the patients achieved a final CMT of less than 240 μm after the treatments; 43% achieved a final CMT of less than 300 μm.

### 3.3. Effects on Intraocular Pressure

Intraocular pressure was determined at baseline, one week after treatment, and at 4 and 6 months after treatment, and the information is presented in Table 7 and Figure 3. In the first study group, four patients experienced raised intraocular pressure at one week after Ozurdex injection, and in the second group, only two patients were found to have high intraocular pressure. For the first group, the baseline mean IOP was 13.9 mmHg, and it was 20.3 mmHg one week after treatment. For the second group (cortisone and anti-VEGF), the baseline mean IOP was 15.4 mmHg, and it was 16.3 mmHg one week after treatment. After the treatment was initiated for patients experiencing raised intraocular pressure, the IOP remained steady throughout the entire follow-up period.

Comparing the IOP values at the one-week follow-up showed a difference of approximately 20% between the two groups. The Ozurdex group had a higher mean value of 20.3 mmHg versus the combined therapy group, which had a mean value of 16.3 mmHg.

### 3.4. Statistical Analysis of the Results

For all three study outcomes, repeated-measures ANOVA was employed to assess whether there were statistically significant effects between treatment groups. By performing post-hoc Tukey comparisons for all three study results, both treatments showed statistical significance between baseline and the follow-up time points.

The interaction between BCVA (LogMAR) and treatment is statistically significant (*p* = 0.009); the results are presented in Table 2. For both groups, macular edema decreased significantly between baseline and 4 and 6 months after treatment (*p* < 0.001).

For the Ozurdex group, there were no statistically significant changes between the 4- and 6-month follow-up. For the cortisone + anti-VEGF group, there were statistically significant changes between the 4- and 6-month follow-ups (*p* < 0.001). Regarding IOP, for the Ozurdex group, there were statistically important changes between baseline and one-week follow-up (*p* < 0.004), but for the cortisone + anti-VEGF group, there were no differences (*p* = 0.321).

## 4. Discussion

Most of the studies found in the literature compare different anti-VEGF agents or Ozurdex with anti-VEGF agents in terms of reducing macular edema and improving visual acuity [30,31]. Less information is found regarding the comparison between Ozurdex and the combined therapy of anti-VEGF with Vitreal S. Therefore, this comparison may be explored more.

This study aimed to evaluate and compare the effectiveness of intravitreal dexamethasone (DEX) implant versus a combined therapy of intravitreal anti-VEGF and cortisone treatments for macular edema due to branch retinal vein occlusion. The effectiveness was estimated using best-corrected visual acuity, central retinal thickness for macular edema, and intraocular pressure (IOP).

An increased level of anti-VEGF leads to ME, so it is important to maintain retinal perfusion to obtain better visual results. Several studies show that both anti-VEGF treatment and Ozurdex represent an effective therapy for retinal vein occlusion-related ME [32,33].

Faye H et al. published a 3-year follow-up study on combination therapy for macular edema in retinal vein occlusions. The study reported 66 retinal vein occlusion patients with ME treated with combination therapy (initially Ranibizumab and later the optional addition of Ozurdex and laser). Visual acuity (VA) and central retinal thickness (CRT) were assessed at baseline and after 1 year. The authors report the following results: baseline LogMAR VA of 0.71 improved to 0.48 at Year 3 (*p* = 0.006); 63% experienced VA improvement (40% improved ≥ 3 lines); and 27% had worse vision. Statistically significant CRT improvement was noted in each year (Year 3 median CRT = 264 µm) compared to baseline (median CRT = 531 µm). There was a reduction in the mean number of total injections to 2.5 in Year 3 (vs. 5.5 in Year 1) [34]. The BCVA values that resulted from our study for the two study groups showed that patients from the cortisone + anti-VEGF group gained one line more on the reading chart than the Ozurdex group. There were no cases of visual acuity decrease among the study subjects at the final follow-up. In our study, for the Ozurdex group, the mean baseline CMT was 427 μm 427 ± 76.8 μm (min. 320 μm, max. 570 μm), which decreased to 347 μm ± 97.9 μm (min. 260 μm, max. 520 μm) at the final follow-up. For the second group, cortisone + anti-VEGF, the mean initial CMT was 471 ± 48.8 μm (min. 394 μm, max. 534 μm), and at the final follow-up, the values improved significantly to 344 ± 58.0 μm (min. 268 μm, max. 480 μm). The data obtained from our study are consistent with those presented in the abovementioned research.

Another reference study had the purpose of assessing various combination treatment options to improve the short-term efficacy of intravitreal monotherapy for the treatment of macular oedema, secondary to retinal vein occlusion for a 12-month follow-up period. This study employed Ranibizumab and Ozurdex with laser photocoagulation treatment. Initial anti-VEGF therapy was performed, controlling recurrent non-ischemic ME, and then an intravitreal steroid injection was applied combined with laser therapy to non-perfused retina. A total of 66 eyes were included in the study. The baseline visual acuity and central retinal thickness were analyzed. At 12 months, 77% had significant VA improvement, 52% had ≥3-line improvement, and 15% were worse. Significant improvements in central retinal thickness were observed: 97% (baseline median CRT = 531 μm (IQR 435–622) reduced to 245 μm (IQR 221–351, *p* < 0.001) at 12 months) and 76% achieved a dry fovea at 1 year. The mean number of total injections required was 5.5 (range 2–11), and 6% required ≥9 injections in 1 year. Although 70% received additional Ozurdex, 82% received ≥1 session of laser therapy [35]. Laser photocoagulation treatment was not performed on our study subjects, but intraocular therapy with Ozurdex, Eylea, and Vitreal S showed significant effects on visual acuity improvement and macular edema reduction.

A large number of studies have proven that intravitreal steroid injections, both Ozurdex and Vitreal S, are effective for treating BRVO macular edema. The effectiveness of such steroid injections is due to their anti-inflammatory properties and their ability to inhibit the release of VEGF. Previous studies have shown that Ozurdex is a well-tolerated, successful treatment [36,37,38,39,40]. The results obtained in this study confirm the effectiveness of both types of intravitreal steroid injections, but the combination of cortisone with anti-VEGF offered a slight improvement in BCVA, by one line. The effects of the therapy were stable between the 4- and 6-month follow-up.

In the GENEVA study, the BCVA results suggested that the peak efficacy was noted at day 60, whereas by day 180, the BCVA results of the Ozurdex-treated patients were no longer significantly better than those of the control group [41,42]. Moreover, the mean change in CRT from the baseline was significantly better on day 90, but not on day 180. In the present study, the mean final change in CMT after the treatments, at 180 days, was 80 μm for the Ozurdex group. For the combined therapy group, the mean final change in CMT was 127 μm. In our study, the peak BCVA and CMT values were reached at 120 days and remained consistent up to 180 days for both treatment groups

Nonetheless, some patients do eventually require retreatment with Ozurdex and anti-VEGF due to recurrent ME. Thus, combined treatment may be an optimal therapeutic solution for retinal vein occlusion-related ME to reduce the injection frequency of anti-VEGF agents and the risk of cataract or ocular hypertension caused by Ozurdex implant [43]. As such, this study was conducted to clarify the effects and effectiveness of Ozurdex in comparison with three intravitreal injections (two with anti-VEGF and one with cortisone) treating BRVO-ME for a period of 6 months after treatment, thereby helping ophthalmologists to choose the best treatment scheme for ME due to retinal vein occlusions.

The possible complications of intravitreal injection include: corneal abrasions and surface irritation (subconjunctival hemorrhages), cataract formation, retinal tear and detachment, vitreous hemorrhage, intraocular pressure elevation, intraocular inflammation, and endophthalmitis. Rates of raised IOP have been estimated at 30% to 60% following an injection of triamcinolone acetonide, and between 30% and 50% after dexamethasone implant. DEX might cause extreme uncontrollable pressure (60 mmHg to 70 mmHg), and sometimes the only solution is vitrectomy. The incidence of traumatic cataract is estimated at 0% to 0.8% [44].

Four subjects from the Ozurdex group, representing 26.66%, and two from the other study group, representing 13.33%, experienced raised intraocular pressure. Additionally, 20% of the total number of studied eyes had raised IOP one week after the intraocular injection was performed. Our study rates for this complication are slightly lower than those mentioned in the above study, but no extreme values were registered—the highest value was 28 mmHg in the Ozurdex group.

A very complex and large research performed on 5318 patients, who received forty-four thousand seven hundred thirty-four injections, established that the complication rates were low, representing 1.9% of all injections, with 1031 unique complications in 685 patients (12.9%). The most common minor complications were irritation (n = 312) and subconjunctival hemorrhage (n = 284). The most frequent serious complication, requiring intervention, was corneal abrasion (n = 46) and iritis (n = 31). Most complications (66%) were managed adequately by a telephone or electronic message [45].

Another minor common complication, also reported in our research, was subconjunctival hemorrhage. Three patients presented subconjunctival hemorrhage: two from the Ozurdex group and one from the combined therapy group. All these patients were already under treatment with artificial tears, and no other treatment was recommended. Furthermore, 10% of the patients presented this complication, and this value was higher than the one reported in the study described above, which had a rate of 5.34% for this complication. Two patients reported conjunctival irritation that lasted for 2–3 days after the treatment.

Vitreous hemorrhage may appear due to intravitreal injection in 0.02% to 4.5% of cases. Arevalo et al. reported a prevalence of tractional retinal detachment in 5.2% of 211 eyes due to intravitreal injections. A large retrospective review calculated the rate of sterile endophthalmitis with aflibercept at 0.16%, compared to 0.10% for Bevacizumab and 0.02% for Ranibizumab. Endophthalmitis after injection is rare, with incidence ranging from 0.01% to 0.3% of cases [46,47].

We report no serious complications that needed surgery or additional treatment, probably due to the relatively small study size.

Intraocular injection complications may occur due to the host reaction or technique or may be drug-induced. Research proves that the most frequent complications are minor and do not require special treatment, and the majority resolve themselves spontaneously. Considering these remarks, we may state that performing three intraocular injections versus one should not lead to additional risks or complications.

The results published in the literature as well as the outcomes of this research prove that both of these therapies can achieve significant functional and anatomical improvements during the therapeutic processes, with no significant differences between these two groups.

## 5. Conclusions

All the patients included in the study were treatment-naïve, and the data were collected under strict criteria. The study design factors ensured that the results were highly accurate.

In this study, the BCVA results suggested that the peak efficacy was at 4 months for both groups. Regarding the CMT, there were no significant changes between the 4- and 6-month follow-up periods, but none of the patients obtained dry retina, meaning that the CMT was less than 240 μm. One of the most frequent complications associated with intravitreal therapy for retinal diseases was elevated IOP [48]. The results of this study indicate that IOP elevation was much higher in the DEX group than that in the cortisone + anti-VEGF group, one week after initiating therapy. All the cases were controlled with topical medication. Such elevations were transient, and all of the patients’ IOPs returned to normal values one week after local treatment with dorzolamide/timolol 20 mg/mL + 5 mg/mL was initiated. In order to prevent secondary IOP spikes, treatment was administered for 6 months. IOP was determined at each follow-up visit, and values were normal. Besides raised IOP, eight patients presented subconjunctival hemorrhages, which required no treatment and disappeared in around 10–14 days.

This study has some potential limitations, including the relatively small sample size, short follow-up period, and absence of a control group. More long-term research with comprehensive outcomes is needed to evaluate the safety and effectiveness of treatment for ME due to retinal vein occlusions. The multifactorial pathogenesis of this disease and treatment of the risk factors (hypertension, diabetes mellitus, and hematological disorders) also require investigation The visual loss resulting from BRVO is due to ME, macular nonperfusion, and retinal neovascularization.

In conclusion, this study confirms that both Ozurdex and the combination of anti-VEGF with cortisone result in significant ME decreases and BCVA improvements in the treatment of BRVO patients.

The International Classification of Diseases 11 (2018) classifies distance vision impairment into four categories. At baseline, our study subjects were considered moderately visually impaired, and after the intraocular treatment was performed, they could be classified into the mild category of visual impairment.

All in all, several studies have reached similar conclusions, but this study highlights the benefits of the combination of anti-VEGF and cortisone (Vitreal S), a much more affordable solution with similar results to the Ozurdex therapy. The financial aspect is a key factor in patients’ compliance to treatment in countries where patients have out-of-pocket expenses for intraocular injections.

## Figures and Tables

**Figure 1 healthcare-11-01414-f001:**
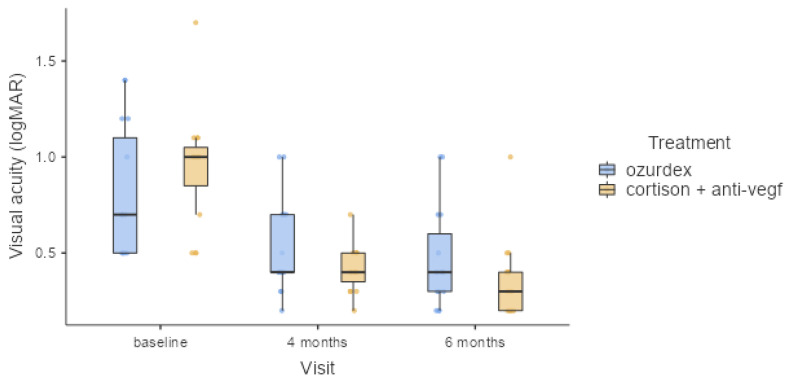
Visual acuity fluctuation at three time points (at baseline and 4 and 6 months after treatment).

**Figure 2 healthcare-11-01414-f002:**
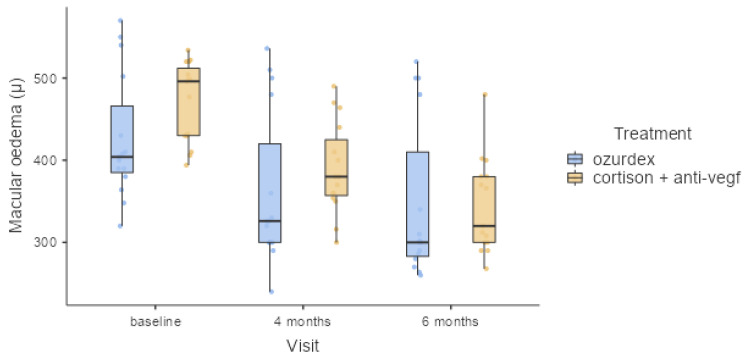
Macular edema variation at three time points (at baseline and 4 and 6 months after treatment).

**Figure 3 healthcare-11-01414-f003:**
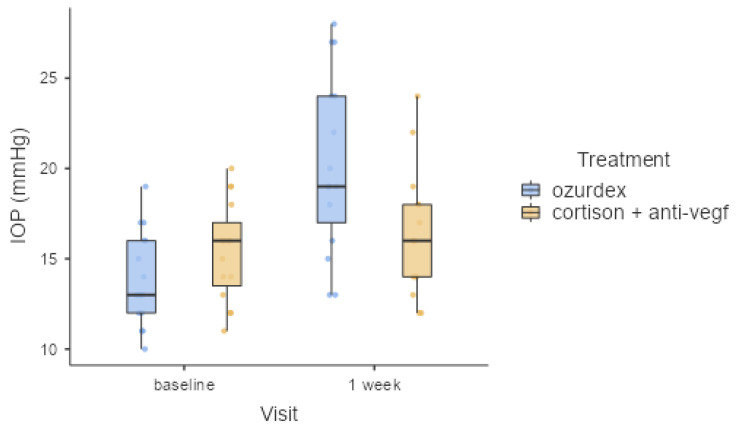
IOP variation from baseline to 1 week after treatment.

**Table 1 healthcare-11-01414-t001:** Patient demographics at baseline.

Parameter	Ozurdex Group	Cortisone + Anti-VEGF Group
Age (average ± SD, years)	72 ± 4.86	67 ± 3.82
Female	7	6
Male	8	9
Right eye	10	8
Left eye	5	7
Mean BCVA baseline(LogMAR)	0.833	0.947
IOP (average ± SD, mmHg)	13.9 ± 2.63	15.4 ± 2.77
Central Retinal Thickness (mean ± SD, μm)	427 ± 76.8	471 ± 48.8
Phakic	3	6
Pseudophakic	12	9

**Table 2 healthcare-11-01414-t002:** Comparison of visual acuity fluctuation at three time points (at baseline and 4 and 6 months after treatment).

	Visit	Treatment	N	Missing	Mean	Median	SD	Min	Max
BCVA (LogMAR)	baseline	Ozurdex	15	0	0.833	0.700	0.337	0.500	1.400
		Cortisone + anti-VEGF	15	0	0.947	1.000	0.307	0.500	1.700
	4 months	Ozurdex	15	0	0.540	0.400	0.247	0.200	1.000
		Cortisone + anti-VEGF	15	0	0.413	0.400	0.119	0.200	0.700
	6 months	Ozurdex	15	0	0.473	0.400	0.266	0.200	1.000
		Cortisone + anti-VEGF	15	0	0.367	0.300	0.206	0.200	1.000

**Table 3 healthcare-11-01414-t003:** Repeated-measures ANOVA—within subject effects.

	Sum of Squares	df	Mean Square	F	*p*
BCVA(LogMAR)	0.00122	2	6.11 × 10^−4^	0.0240	0.976
BCVA(LogMAR)—Treatment	0.26185	2	0.13092	5.1464	0.009
BCVA(LogMAR)—Age	0.01735	2	0.00868	0.3410	0.713
Residual	1.37376	54	0.02544		

Note. Type 3 Sums of Squares.

**Table 4 healthcare-11-01414-t004:** Repeated-measures ANOVA—between subject effects.

	Sum of Squares	df	Mean Square	F	*p*
Treatment	2.07 × 10^−4^	1	2.07 × 10^−4^	0.00138	0.971
Age	0.0881	1	0.0881	0.58488	0.451
Residual	4.0674	27	0.1506		

Note. Type 3 Sum of Squares.

**Table 5 healthcare-11-01414-t005:** Post-Hoc Comparisons—Treatment.

Comparison						
Treatment	Treatment	Mean Difference	SE	df	t	Ptukey
Ozurdex	Cortisone + anti-VEGF	0.00351	0.0947	27.0	0.0371	0.971

**Table 6 healthcare-11-01414-t006:** Comparison of macular edema variation at three time points (at baseline and 4 and 6 months after treatment).

	Visit	Treatment	N	Mean	Median	SD	Min	Max
Macular edema (μm)	baseline	Ozurdex	15	427	404	76.8	320	570
		Cortisone + anti-VEGF	15	471	496	48.8	394	534
	4 months	Ozurdex	15	361	326	94.4	240	536
		Cortisone + anti-VEGF	15	390	380	55.9	300	490
	6 months	Ozurdex	15	347	300	97.9	260	520
		Cortisone + anti-VEGF	15	344	320	58.0	268	480

**Table 7 healthcare-11-01414-t007:** IOP fluctuation at baseline and 1 week after treatment.

	Visit	Treatment	N	Mean	Median	SD	Min	Max
IOP (mmHg)	Baseline	Ozurdex	15	13.9	13	2.63	10	19
		Cortisone + anti-VEGF	15	15.4	16	2.77	11	20
	1 week	Ozurdex	15	20.3	19	4.95	13	28
		Cortisone + anti-VEGF	15	16.3	16	3.48	12	24

## Data Availability

Data shared are in accordance with consent provided by participants on the use of confidential data. Data sharing is not applicable to this article.

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
