# Peer review of "The Impact of Intraocular Treatment on Visual Acuity of Patients Diagnosed with Branch Retinal Vein Occlusions"

_healthcare, 2023, doi:10.3390/healthcare11101414_

Round 1
Reviewer 1 Report
In this paper, the authors tried to compare the effects of Ozurdex and anti-VEGF plus cortisone combination therapy in treatment of BRVO-ME in 30 eyes. The visual acuity, macular edema and intraocular pressure (IOP) were evaluated at 4- and 6-month after injection. There are several issues should be addressed.
1. Table 1 showed patient demographics of all involved cases. It will be better to compare age, BCVA baseline, central retinal thickness and IOP of the two groups, to confirm that there is no significant difference before treatment.
2. The authors should compare BCVA, central retinal thickness and IOP at all time points, to see if the two therapy had any different affects.
3. The authors should describe complications during the therapy or associated with intravitreal injections. For the six patients who raised intraocular pressure after treatment, how long did their IOP return to normal?
Author Response
Dear reviewer,
Thank you very much for your assessment.
Regarding your observations, we tried to fulfill all of them.
- Table 1 showed patient demographics of all involved cases. It will be better to compare age, BCVA baseline, central retinal thickness and IOP of the two groups, to confirm that there is no significant difference before treatment.
At baseline, there were no important differences between the studied parameters, data is presented in table 1.
We added new information in the demographics table, on groups, age, BCVA, IOP, CMT.
- The authors should compare BCVA, central retinal thickness and IOP at all time points, to see if the two therapy had any different affects.
We added the information in each results section, BCVA, CMT and IOP
Comparing the BCVA values, between the two study groups at 4 and 6 months follow up, showed that there were no major differences, not more than one line on the reading chart. All values fluctuations are presented in table 2.
Comparing the CMT values, between the two study groups at 4 and 6 months follow up, proved that there were no significant differences. At 4 months a mean value of 31 μm of difference between the two groups and at 4 months the mean difference was less the 3 μm . All values fluctuations are presented in table 6.
Comparing the IOP values at 1 week follow up , showed a difference of approximately 20% between the 2 groups. The Ozurdex group had a higher mean value of 20.3 mmHg versus the combined therapy group which had a mean value of 16.3 mmHg.
- The authors should describe complications during the therapy or associated with intravitreal injections. For the six patients who raised intraocular pressure after treatment, how long did their IOP return to normal?
We added the information in the conclusions part:
Patients’ IOPs returned to normal values, one week after local treatment with dorzolamide/timolol 20mg/ml + 5 mg/ml, was initiated. In order to prevent secondary IOP spikes, treatment was administered for 6 months. IOP was determined at each follow up visit, and values were normal.
And in the results section :
“After the treatment was initiated for patients experiencing raised intraocular pressure, the IOP remained steady throughout the entire follow up period.”
We discussed more profound the possible complications.
The possible complications of intravitreal injection include: corneal abrasions and surface irritation (subconjunctival hemorrhages ), cataract formation, retinal tear and detachment, vitreous hemorrhage, intraocular pressure elevation, intraocular inflammation, and endophthalmitis. Rates of raised IOP have been estimated at 30% to 60% following an injection of triamcinolone acetonide, and between 30% and 50% after dexamethasone implant. DEX might cause extreme uncontrollable pressure ( 60 mmHg to 70 mmHg) and sometimes the only solution is vitrectomy. The incidence of traumatic cataract is estimated at 0% to 0.8%. 44
A very complex and large research performed on 5318 patients, whom received forty-four thousand seven hundred thirty-four injections, established that the complication rates were low, representing 1.9% of all injections, with 1031 unique complications in 685 patients (12.9%). The most common minor complications, were irritation (n = 312) and subconjunctival hemorrhage (n = 284). The most frequent serious complications, requiring intervention, were corneal abrasion (n = 46) and iritis (n = 31). Most complications (66%) were managed adequately by a telephone or electronic message. 45
Vitreous hemorrhage may appear due to intravitreal injection in 0.02% to 4.5% of cases. Arevalo et al reported a prevalence of tractional retinal detachment in 5.2% of 211 eyes due to intravitreal injections. A large retrospective review calculated the rate of sterile endophthalmitis with aflibercept at 0.16%, compared to 0.10% for bevacizumab and 0.02% for ranibizumab. Endophthalmitis after injection is rare, with incidence ranging from 0.01% to 0.3% of cases. 46,47
Intraocular injections complications may occur due to the host reaction, drug induced or the technique. Researches prove that the most frequent complications are minor and do not require special treatment, the majority resolve themselves spontaneously. Considering this remarks, we may state that performing three intraocular injections versus one, should not lead to additional risks or complications.
Kind regards,
Alexandra Preda
Reviewer 2 Report
Congratulations! This is a very good manuscript and work. I do not have much to tell you about it. Only to considerations:
Firstly, I have read macular edema and macular oedema. Is it a mistake?
On the other hand, I have missed information about Statistical Analysis of the Results in methods. I have read this information after in results.
Author Response
Dear reviewer,
Thank you very much for your kind evaluation.
As we saw in other studies both terms, edema and oedema are used, but thank you for your remark, we corrected, and left edema.
We also mentioned a phrase about the statistical tests in methods section.
Kind regards,
Alexandra Preda
Reviewer 3 Report
The article titled, "The Impact of Intraocular Treatment on Visual Acuity of Patients Diagnosed with Branch Retinal Vein Occlusion" by Diana-Maria Darabus et al., is a valuable contribution to the field of retinal vein occlusion diagnosis and treatment. However, before being considered for publication, the Authors should address the following points:
1-The Authors, in both the abstract and conclusion, state that the combination of anti-VEGF and cortisone in the treatment of BRVO-ME is a more affordable solution when compared to Ozurdex therapy. The Authors should better address this statement and clarify the meaning of "more affordable" in this context.
2-IOP was measured two times: before treatment and a week after the completion of treatment for both groups. Why was the IOP not measured during treatment?
3-The Authors should better address why, in their opinion, the combined therapy is better than the Ozurdex treatment, considering that the combined therapy requires three intraocular injections, while the Ozurdex treatment requires only one injection. Clinical complications related to the two different therapeutic regimens should be discussed in depth.
3-All non-standard abbreviations/acronyms should be written out in full when first used (in both the abstract and the paper itself) and followed by the abbreviated form in parentheses.
4-The manuscript requires a thorough English revision as a few spelling and syntax errors are present.
Author Response
Dear reviewer,
Thank you very much for your evaluation and recommendations. We tried to clarify and added the requested information.
1-The Authors, in both the abstract and conclusion, state that the combination of anti-VEGF and cortisone in the treatment of BRVO-ME is a more affordable solution when compared to Ozurdex therapy. The Authors should better address this statement and clarify the meaning of "more affordable" in this context.
We tried to clarify this expression, in our country patients pay for the intraocular treatment. For almost a year now, Eylea and Beovu are for free, but Vitreal and Ozurdex must be paid…
The financial aspect is a key factor in patients’ compliance to treatment in countries where patients have out-of-pocket expenses for intraocular injections.
2-IOP was measured two times: before treatment and a week after the completion of treatment for both groups. Why was the IOP not measured during treatment?
I apologize for not being so clear about this. We added this clarification in the conclusion part;
“Patients’ IOPs returned to normal values, one week after local treatment with dorzolamide/timolol 20mg/ml + 5 mg/ml, was initiated. In order to prevent secondary IOP spikes, treatment was administered for 6 months. IOP was determined at each follow up visit, and values were normal.”
And thi in the results section:
“After the treatment was initiated for patients experiencing raised intraocular pressure, the IOP remained steady throughout the entire follow up period.”
3-The Authors should better address why, in their opinion, the combined therapy is better than the Ozurdex treatment, considering that the combined therapy requires three intraocular injections, while the Ozurdex treatment requires only one injection. Clinical complications related to the two different therapeutic regimens should be discussed in depth.
We added this section to the discussion part:
The possible complications of intravitreal injection include: corneal abrasions and surface irritation (subconjunctival hemorrhages ), cataract formation, retinal tear and detachment, vitreous hemorrhage, intraocular pressure elevation, intraocular inflammation, and endophthalmitis. Rates of raised IOP have been estimated at 30% to 60% following an injection of triamcinolone acetonide, and between 30% and 50% after dexamethasone implant. DEX might cause extreme uncontrollable pressure ( 60 mmHg to 70 mmHg) and sometimes the only solution is vitrectomy. The incidence of traumatic cataract is estimated at 0% to 0.8%. 44
A very complex and large research performed on 5318 patients, whom received forty-four thousand seven hundred thirty-four injections, established that the complication rates were low, representing 1.9% of all injections, with 1031 unique complications in 685 patients (12.9%). The most common minor complications, were irritation (n = 312) and subconjunctival hemorrhage (n = 284). The most frequent serious complications, requiring intervention, were corneal abrasion (n = 46) and iritis (n = 31). Most complications (66%) were managed adequately by a telephone or electronic message. 45
Vitreous hemorrhage may appear due to intravitreal injection in 0.02% to 4.5% of cases. Arevalo et al reported a prevalence of tractional retinal detachment in 5.2% of 211 eyes due to intravitreal injections. A large retrospective review calculated the rate of sterile endophthalmitis with aflibercept at 0.16%, compared to 0.10% for bevacizumab and 0.02% for ranibizumab. Endophthalmitis after injection is rare, with incidence ranging from 0.01% to 0.3% of cases. 46,47
Intraocular injections complications may occur due to the host reaction, drug induced or technique. Researches prove that the most frequent complications are minor and do not require special treatment, the majority resolve themselves spontaneously. Considering this remarks, we may state that performing three intraocular injections versus one, should not lead to additional risks or complications.
3-All non-standard abbreviations/acronyms should be written out in full when first used (in both the abstract and the paper itself) and followed by the abbreviated form in parentheses.
We checked and corrected them.
4-The manuscript requires a thorough English revision as a few spelling and syntax errors are present.
We made some changes, but the manuscript was corrected and edited by mdpi services. I will attach the certificate, but if you consider that more changes should be done, I will rewrite to them.
Kind regards,
Alexandra Preda
Round 2
Reviewer 1 Report
In their revised manuscript, the authors have responded to the comments provided in the original reviews. They also included more discussions for helping comprehensively explain the results. But there are still some issues need to be addressed.
1. In Table 1, it will be better to show the two groups in two different columns.
2. The authors added possible complications in the discussion. But they should focus more on complications happened (or not happen) in this study, and compare with previous studys.
Author Response
Dear reviewer,
Thank you very much for your suggestions and corrections.
- As you recommended, we changed table 1, on columns.
-
- The authors added possible complications in the discussion. But they should focus more on complications happened (or not happen) in this study, and compare with previous studys.
We compared our complications and their rate with the one mentioned in the discussed studies. We added this paragraphs:
4 subjects from the Ozurdex group, representing 26,66% and 2 from the other study group, representing 13,33%, experienced raised intraocular pressure. 20% of the total number of studied eyes, had raised IOP, one week after the intraocular injection was performed. Our study rates for this complication are slightly lower than those mentioned in the above study, but no extreme values were registered, the highest value was 28 mmHg in the Ozurdex group.
Another minor common complication, also reported in our research, was subconjunctival hemorrhage. 3 patients presented subconjunctival hemorrhage, 2 from the Ozurdex group and 1 from the combined therapy group. All this patients, were already under treatment with artificial tears, and no other treatment was recommended. 10% of the patients presented this complication, this value being higher than the one reported in the above described study, which had a 5,34% rate for this complication. 2 patients reported conjunctival irritation, that lasted for 2-3 days after the treatment.
We report no serious complication, that needed surgery or additional treatment, probably due to the relative small study lot.
Kind regards,
Alexandra Preda